# Perceiving the arrow of time in autoregressive motion

**Kristof Meding**
University of Tübingen
Neural Information Processing Group
Tübingen, Germany
kristof.meding@uni-tuebingen.de

**Dominik Janzing**
Amazon Research Tübingen
Tübingen, Germany
janzind@amazon.com

**Bernhard Schölkopf\***
Max-Planck-Institute for Intelligent Systems
Empirical Inference Department
Tübingen, Germany
bs@tuebingen.mpg.de

**Felix A. Wichmann\***
University of Tübingen
Neural Information Processing Group
Tübingen, Germany
felix.wichmann@uni-tuebingen.de

*joint senior authors

## Abstract

Understanding the principles of causal inference in the visual system has a long history at least since the seminal studies by Albert Michotte. Many cognitive and machine learning scientists believe that intelligent behavior requires agents to possess causal models of the world. Recent ML algorithms exploit the dependence structure of additive noise terms for inferring causal structures from observational data, e.g. to detect the direction of time series; the arrow of time. This raises the question whether the subtle asymmetries between the time directions can also be perceived by humans. Here we show that human observers can indeed discriminate forward and backward autoregressive motion with non-Gaussian additive independent noise, i.e. they appear sensitive to subtle asymmetries between the time directions. We employ a so-called frozen noise paradigm enabling us to compare human performance with four different algorithms on a trial-by-trial basis: A causal inference algorithm exploiting the dependence structure of additive noise terms, a neurally inspired network, a Bayesian ideal observer model as well as a simple heuristic. Our results suggest that all human observers use similar cues or strategies to solve the arrow of time motion discrimination task, but the human algorithm is significantly different from the three machine algorithms we compared it to. In fact, our simple heuristic appears most similar to our human observers.

## 1 Introduction

Discriminative convolutional neural networks (CNNs) have produced impressive results in machine learning, but certain striking failures of generalisation have been pointed out as well in terms of adversarial examples [1–3] or the recent findings of Geirhos and colleagues that CNNs show surprisingly large generalisation errors under image degradations [4, 5]. Many cognitive and machine learning scientists maintain that flexible and robust intelligent behaviour *in the real world* requires agents to possess generative or causal models of the world [6]. The importance of causality for cognitive science and psychology has long been recognized [7–16]. In visual perception, for example, it is fundamental to identify the causal structure in a visual scene: are objects moving or standing still, are some objects causing the movement of other objects [17–19], are the movements caused by (intentional) actors or rather by forces of nature? [20, 21] On a cognitive rather than perceptual level progress has been made to understand how we intuitively understand physics [22], how humans learn

causal structures from data [10, 7, 8, 13] and on human causal inference via counterfactual reasoning [16, 23].

Much less research has explored whether the earlier, perceptual and unconscious—cognitively impenetrable [24]—processing stages in humans possess already causal inference algorithms, see Danks [25] for a recent overview on the relationship between causal perception, causal inference and causal reasoning. Rolfs et al. [26] found evidence for perceptual adaptation to causality, thus arguing that the perceptual system already possesses mechanisms tuned to "causal features" in the visual input (but c.f. for a critique of the paper on methodological grounds [27]). More recently it was shown, using the continuous flash suppression paradigm, that simple Michotte style launching-events enter awareness faster when they are perceived as continuous causal events, again suggesting that rather early, perceptual and pre-conscious processes may already be tuned to "causal features" [28].

Recently there has been considerable progress in understanding causal inference by approaching it as a machine learning problem [29–32]. In the last two decades algorithms for causal inference with different approaches have been suggested. Based on the language of graphical models and structural equation models, the "classical approaches" infer the directed acyclic graph (DAG) formalizing the causal relations from the observed conditional statistical (in)dependences subject to causal Markov condition and causal faithfulness [29, 30]. After about 2004, several other approaches were suggested that infer causal DAGs using properties of distributions other than conditional independence. These approaches also consider DAGs that consist of two variables only (in which case conditional independence testing is futile), i.e., to decide what is cause and what is effect, see chapter 4 of [32] for an overview. It was shown that one can still infer the structure if one is willing to place restrictions on the action of the noise disturbances, specifically, that it is additive and independent, and that either the noise is non-Gaussian or the functions are nonlinear [33–36]. These methods have also been applied to determine the causal direction of time series by fitting autoregressive models, i.e., by predicting future from past, and examining the noise terms [37–39].

Investigation of the arrow of time in causal learning was motivated by its role in physics [40, 41, 37], since it can be shown that the time asymmetry based on the independence of noise can be explained by the usual thermodynamic arrow of time [42] and that recent approaches to causal inference are thus linked to statistical physics [43]. Pickup et al. showed that the independence of noise can be employed to detect the arrow of time in real world YouTube videos, without semantic or cognitive knowledge about the visual world [39]. Recently it was shown that also neural networks can infer the arrow of time from movies alone [44], suggesting that even low-level motion information in the video contains information about the arrow of time.

Clearly, humans can perceive the arrow of time in settings where semantic information or world knowledge is available. In a famous movie by the Lumière brothers, a wall falls over, subsequently shown backwards to illustrate the perceptual contrast.[1] Similarly, humans can perceive the arrow of time if there is a clear non-stationarity in the data, or a directionality due to a perceivable increase in entropy, e.g. if we observe an explosion. However, ML causality methods can also infer the arrow of time in cases that at first sight appear hard, i.e. where the marginals are the same in both motion directions and the setting is stationary. For humans, in contrast, the perception of the arrow of time in such settings is unclear. Although it is well established that humans are sensitive to higher-order regularities in the *spatial* statistics of static natural images [45], for motion sequences or motion discrimination analogous results have not yet been established. It was even recently shown—at least when assessing the motion direction of random dot kinematograms (RDKs)—that humans appear only sensitive to the mean and variance of the displacement angles but were insensitive to skewness and kurtosis [46]. Thus, for RDKs, and unlike in the case of static spatial structure, the human visual system appears insensitive to higher-order statistics. Causal dependency algorithms, however, in the linear case crucially rely on non-Gaussianity of additive noise, for which kurtosis and additional non-zero higher-order moments are a measure.

Thus we investigated whether the human visual system is sensitive to dependencies in the motion of a single disk. Furthermore, we investigate in depth the relationship between the abilities of different machine learning algorithms: a Residual Dependence based algorithm, a Neural Network, a Bayesian ideal observer and a very simple ecological valid heuristic. We show, first, that human observers can indeed discriminate the arrow of time in autoregressive (AR) motion with non-Gaussian additive independent noise, i.e. they appear sensitive to subtle time reflection asymmetries. Second, we show

that humans are remarkably efficient in this task, requiring only a short motion sequence to identify the direction of the time series. Third, humans might use a strategy similar to the heuristic. Fourth, we show that the ideal observer algorithm and the neural network both achieve "super-human"—and quantitatively very similar—performance, but the frozen noise paradigm we employed shows that both algorithms use different cues or strategies.

## 2 Methods

Here we provide the minimum information necessary to understand our experiments and results. We refer to the supplementary material for detailed explanations and all information needed to allow all experiments to be reproduced.

### 2.1 The arrow of time: Causal and anti-causal time series

We constructed time series from a generative additive noise model:

$$x_t = 0.05 \cdot x_{t-4} + 0.1 \cdot x_{t-3} + 0.2 \cdot x_{t-2} + 0.4 \cdot x_{t-1} + \epsilon_t$$

The noise $\epsilon_t$ is independent from all previous states $x_{t-1}, x_{t-2}, \ldots$. Clearly, future states $x_t, x_{t+1}, x_{t+2}, \ldots$ are dependent on $\epsilon_t$ since $\epsilon_t$ influences them (the arrow of time in this setting). This is true for all types of noise distributions for $\epsilon_t$, however, the direction is not detectable for Gaussian noise in a linear time series since a linear Gaussian time series can be modeled in the forward and backward direction with independent noise terms. For non-Gaussian noise, however, this is not true: it is not possible to fit a time series in the backward direction with independent noise terms [37].

Multiple algorithmic ways exist to detect the direction of such a time series based on this dependence structure. We describe them in section 2.3. Note that we can use the case with the Gaussian distribution for $\epsilon_t$ as "sanity check" to test our psychophysical experiment as well as our algorithms: neither humans nor algorithms should be able to identify the direction with Gaussian noise.

Throughout we use time series for which the additive noise component $\epsilon_t$ is distributed according to $\epsilon_t \sim sgn(Y) \cdot |Y|^r$, with $Y$ Gaussian distributed. We choose the exponent $r$ in the range of $0.1 - 6$. This yields noise which is either Bimodal ($r < 1$) or peaked Super-Gaussian ($r > 1$). The closer the value of $r$ is to 1, the more Gaussian $\epsilon_t$ becomes. An Exponent $r = 1$ yields Gaussian distributed noise. The noise parameterization with exponent $r$ has the advantage that the non-Gaussianity of the time series can be precisely controlled with one parameter. Additionally, we choose a single smoothed Uniform distribution with tails extending to $\pm\infty$. In total 16 noise distributions were used in our experiment, seven with Super-Gaussian additive noise, seven with Bimodal additive noise, one with smoothed Uniform and one with Gaussian additive noise. All noise distributions had mean $0$ and standard deviation of $44.72$ pixels on screen (1,13 cm), see appendix sec. A.1. These values ensure, in practice, that the time series is bounded to the range of possible coordinates of the monitor used in our experiment. Time series in the true time direction are in the following denoted as causal time series, and time series which are flipped along the temporal axis are denoted as anti-causal. Movies of the stimuli are presented in the supplementary material.

### 2.2 Psychophysical Experiment

We tested if humans have the ability to discriminate causal from anti-causal time series in a psychophysical experiment. Observers saw a white random dot moving across the horizontal axis on a computer screen. The dot position followed a linear non-Gaussian time series with additive noise described as above. Observers had to press a button whether they saw the moving dot belonging to the green (causal) or to red (anti-causal) category—observers were unaware that the difference between the categories was a time-reversal; they were given a cover story to identify harmless from dangerous bacteria based on their motion. We hypothesized that humans are better at classifying very strong non-Gaussian time series as algorithms do [37]. Thus we began by training subjects with easily classifiable noise and made the time series progressively more difficult (making $r$ approach 1.0). Human observers should be able to use the same cue for different intensities of the Bimodal or Super-Gaussian noise. The discrimination task is rather difficult and we screened participants based on their performance in what we considered "easy conditions" with $r = 6, 4, 2$ (Super-Gaussian) and $r = 0.1, 0.3, 0.5$ (Bimodal). Participants had to achieve at least $67.5\%$ in these blocks (40 trials) to

be significantly different from chance level and to participate further in our experiment. Seven of our 17 naive observers failed to reach the criterion. We provide detailed information in the supplementary material A.2.2 why we think this does not influence our overall results about human performance

Ten naive observers participated successfully in the first experiment (6 female, 4 male mean, mean age = 24 yrs, std = 2.5 yrs). All subjects received monetary compensation. The observers were tested on time series with all 16 noise distributions. For Bimodal and Super-Gaussian noise observers progressed from easy to difficult noise. Each observer classified every of the 16 noise distributions 40 times, 640 trials in total per observer and it took each observer four hours to complete the first experiment.

The first experiment assessed how well observers were able to discriminate forward and backward AR motion sequences as a function of the degree of non-Gaussianity of the additive noise, i.e. to see the arrow of time. Our second experiment aimed to investigate both human and algorithmic strategies for the detection of the arrow of time. In this experiment the noise was randomly sampled for all subjects. To this end all subjects were tested on exactly the same time series: the so-called frozen noise paradigm often successfully employed in auditory psychophysics [47–49] This experimental technique allows to examine inter-subject or subject-algorithmic correlation and consistency. In the second experiment we only used a single noise distribution, Bimodal noise with exponent $r = 0.5$. The length of the motion sequence—and thus the viewing time—was reduced progressively from the initial 100 time-points to finally only 2 time points (100, 50, 25, 20, 16, 12, 8, 4, 2). Participants classified 40 trials for each sequence length yielding in total 360 trials per observer. Similar to experiment one the task got more difficult as the experiment progressed. Four of the best observers in the previous experiment participated in this experiment (2 male, 2 female, age =22.5 yrs, std = 2.3 yrs). The experiment lasted 1.5 hours per observer.

## 2.3 Algorithms for causal inference

One central aim of ours is to compare the abilities of humans and algorithms to detect the arrow of time. We compared the performance of our human observers to three different algorithms: First, an algorithm which directly exploits the residual dependence structure (ResDep). Second, a neurally inspired network and, third, a Bayesian ideal observer algorithm. Furthermore, a simple heuristic is tested.

The ResDep algorithm proposed by Jonas Peters et al. [37] uses directly the residual dependence structure of $\epsilon_t$ to the value $x_{t-1}$. The algorithm fits an autoregressive model to the time series and a series flipped along the time dimension. Subsequently an independence test is performed between fitted residuals and data points. The direction is decided using the Hilbert-Schmidt Independence Criterion test. The true time direction maximizes the independence-score between residuals and data points.

The second algorithm was a (simple) neurally inspired network [50, 51]. The network consisted of one convolution layer, followed by a batch normalization layer, a ReLU-layer and a fully connected layer (see A.3.1 in the appendix for further details). For each noise distribution the network was trained with 30000 time series. We used the Adam optimizer with an initial learning rate of 0.01. The network was trained for a maximum of 30 epochs. Both the ResDep algorithm and the neural network has full temporal memory since we input the full time series at the first step.

While the neural network gets as input the full time series and thus has perfect temporal memory, we can contrast this algorithm with one based on Bayes statistics. In the vision literature this is often done in an ideal observer framework [52]. An ideal observer analysis is a statistical framework which provides the upper limit of performance *given a set of constraints* since the ideal observer has perfect knowledge about the underlying model and its constraints.

We calculated the probability of the direction $d$ given the data $X = (x_1, x_2, ..., x_N)$ using Bayes rule:

$$p(d|X) = \frac{p(X|d) \cdot p(d)}{p(X)} = \frac{\Pi_{t=1}^N p(x_t|x_{t-1}, x_{t-2}, ..., x_1, d) \cdot p(d)}{p(X)}.$$

If we consider only stationary and stable time series of order 4—as in our experiments—the terms in the numerator become $p(x_t|x_{t-1}, x_{t-2}, x_{t-3}, x_{t-4})$ for the forward time series. This term corresponds exactly to the chosen noise distribution. We compare this expression in the forward and backward direction and choose the direction for which the corresponding probability is larger. This method is very similar to calculating the Bayes Factor. See section A.3.1 in the appendix for a detailed explanation.

As a final algorithm we fitted a heuristic to the data in spirit similar to heuristics proposed e.g. by Stengård and Berg [53]. The heuristics were developed after we had evaluated the feedback from our observers and the analysis of the noise structure. We found two different principles for Bimodal and Super-Gaussian noise. For Super-Gaussian noise noise values are often sampled around 0. Therefore in the forward direction, the dot often jumps around the center and rarely makes a big jump outwards. After such a big jump, the point slowly sprints back to the center. This means that in the forward direction there are big jumps to the outside, in the backward direction there are big jumps to the centre. The Bimodal condition behaves the other way around. Often large values are sampled and only rarely smaller ones. We have used this observation to develop a heuristic in a few lines of code. At the maximum displacement, it is checked whether a large jump occurred before or after it. This 5-line code heuristic also works to identify the arrow-of-time of "real" data (EEG recordings; 60% accuracy). For details see section A.3.1 in the appendix.

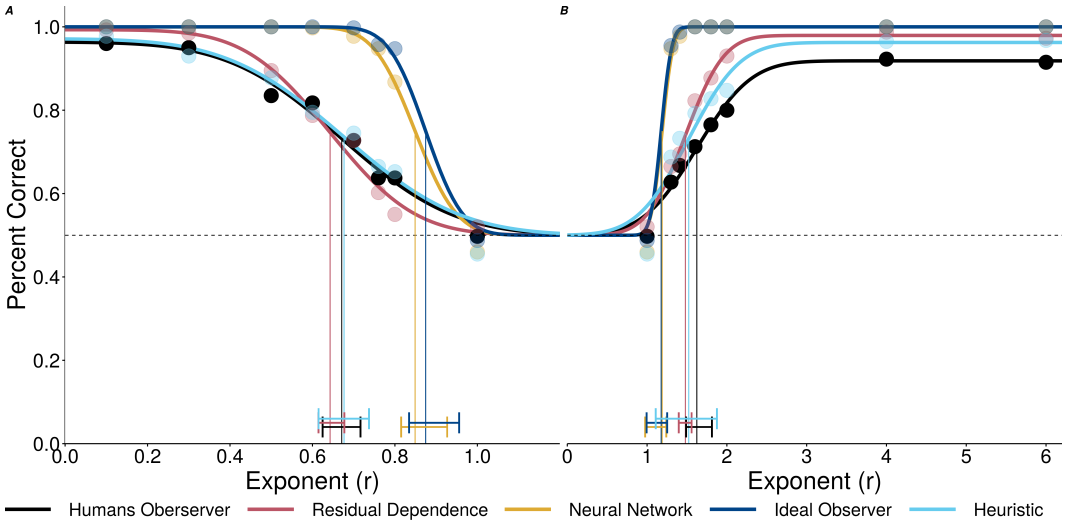

Figure 1: Psychometric Function for Bimodal noise (A) and Super-Gaussian noise (B). The black dots represent the human accuracy for different exponents, pooled over all 10 subjects. The psychometric functions are fitted with cumulative Gaussian distributions. Performance gets worse towards an exponent of 1 which corresponds to the non-identifiable Gaussian noise case. The horizontal line marks the width, the scaled 75% threshold for the different fits. Whiskers show 95% Credible Intervals (CI) for the threshold.

## 3   Results

The following psychometric functions and Bayesian Credible Intervals (CI) were calculated with the Beta Binomial Model in Psignifit 4 [54]. Figure 1 shows the main result of Experiment one. The black psychometric functions show the human data (pooled across all ten observers) and the coloured curves results for the algorithms on the same time series seen by our human observers: ResDep in red, the neural network in yellow, the ideal observer in blue, our heuristic in light blue. Data for single observers are shown in Figure A.8. All individual psychometric functions can be found in figure A.9 and A.10, and the thresholds with credible intervals in table 4. On the one hand, humans can indeed detect the direction of a time series for Super-Gaussian and Bimodal noise with thresholds $r = 0.67, 95\%$ CI $[0.62, 0.72]$ (Bimodal) and $r = 1.62, 95\%$ CI $[1.45, 1.81]$ (Super-Gaussian). The ResDep algorithm, on the other hand, performs similar to humans with Bimodal noise (threshold $r = 0.64, 95\%$ CI $[0.61, 0.68]$) and, perhaps, marginally better with Super-Gaussian noise (threshold $r = 1.48, 95\%$ CI $[1.4, 1.56]$)[2]. Algorithmic performance of the neural network and the ideal observer is superior to human and ResDep performance and both algorithms show remarkably similar results. A detailed analysis of the neural network can be found in A.5. Thresholds for the exponents of the Neural Network are $r = 0.85, 95\%$ CI $[0.82, 0.96]$ and $r = 1.19, 95\%$ CI $[0.98, 1.24]$, for the ideal observer $r = 0.87, 95\%$ CI $[0.83, 0.96]$ and $r = 1.18, 95\%$ CI $[0.99, 1.25]$ and for the heuristic

r $= 0.67, 95\%$ CI $[0.61, 0.74]$ and r $= 1.52, 95\%$ CI $[1.11, 1.88]$.

The parameterization with exponent $r$ is somewhat arbitrary and we tested other distant scales (Kullback-Leibler divergence, Jensen-Shannon divergence, Kolmogorov-Smirnov distance and normalized exponents), see figure A.14. Normalized exponents yield most similar scales.

The results for the smoothed Uniform noise were much more diverse—remember that there is only a single smoothed Uniform distribution with zero mean and the same variance as all other noise distributions we used: The average human accuracy was 50% (chance performance), for ResDep 70%, for the neural network 96%, for the ideal observer 97% and for the heuristic 75%; we discuss these results in the next section.
From Figures 1 and the block by block comparison in A.15 it appears as if human observers may use an internal algorithm similar to ResDep (top left panel in A.15) or the heuristic, and the neural network may have learned a strategy mimicking that of the ideal observer,

The frozen noise paradigm described above in section 2.2 and used in our experiment 2 allows us to investigate this question in a much more stringent way: All human observers and the algorithms classified exactly the same time series—they were not only drawn from the same distribution but the very same time series. In addition, in experiment 2 we explored how human observers and algorithms cope with shorter time series (Bimodal noise, $r = 0.5$ fixed throughout the experiment). This, too, may offer a way to distinguish human observers and algorithms from each other.

Figure 2 shows the results for experiment 2. Plotting conventions as in Figure 1: The black psychometric function shows human data (pooled across all four observers) and the coloured curves results for the algorithms on exactly the same time series seen by our human observers: ResDep in red, the neural network in yellow, the ideal observer in blue, the heuristic in light blue. Individual psychometric functions of the four human observers are shown in Figure A.16. The neural network was exactly trained as in experiment one with the exception that we shrink the size of the convolutional layer for very short time series, see A.3.1 for details. Human observers are able to detect the direction of time series even for rather short time series, with a threshold of about 17.76, 95% CI [14.40,22.44] time points. The results are even more impressive if we exclude observer 2—who told us after the experiment that he had been not fully attentive during the experiment: the threshold drops to 15.17 time points, 95% CI [11.51,19.18], see figure A.17 in the appendix. In this respect, humans clearly outperform the ResDep algorithm which requires 42.67, 95% CI [28.88,58.86] time points for 75% correct discrimination. The neural network with a threshold of 8.13, 95% CI [-1.85,12.52] time points and the ideal observer algorithms with a threshold of 7.73, 95% CI [-0.71,11.16] again show similar performance and are again superior to that of human observers and ResDep. However the heuristic shows a threshold very close to human observers, 18.07 time points, 95% CI [14.20, 46.50]. Please note, however, that the somewhat poor performance of ResDep may not (only) reflect its intrinsic inferiority but may in part be due to the difficulty of fitting short time series. ResDep relies on the ARMA method in MATLAB; ResDep is effectively guessing for time series shorter or equal than 8 time points. Also, the ideal observer has intrinsic problems with short time series since our underlying assumptions for the approximation does not hold anymore, see A.3.1 in the supplementary material for further details.

The frozen noise method allows us to compare observer consistency within observers and consistency between humans and algorithms. If subject 1 has for a given block an accuracy $p_1$ and subject 2 has for the same block an accuracy $p_2$, then we would expect for independent binomial observers a fraction of $p_1 \cdot p_2 + (1 - p_1) \cdot (1 - p_2)$ equally answered ("consistent") trials. This fraction of expected consistency is compared to the number of actually equally answered trials per block. If the observed proportion is significantly higher than expected, this provides evidence that subjects 1 and 2—be them two humans, two algorithms or a human and an algorithm—are not independent, which in turns indicates that they rely on similar processing strategies or at least use similar stimulus information.
Figure 3 shows this comparison for humans and algorithms, with the expected consistency shown on the x-axis, plotted on the observed consistency on the y-axis. Comparing human observers to each other (top left panel in Figure 3), we see that humans tend to have more similarities than expected from independent observers. (The shaded ellipsoidal regions indicate the confidence regions around the null hypothesis that they are independent given the amount of data.) The first column in Figure 3 strongly suggests that humans observers use a strategy or internal algorithm independent from all

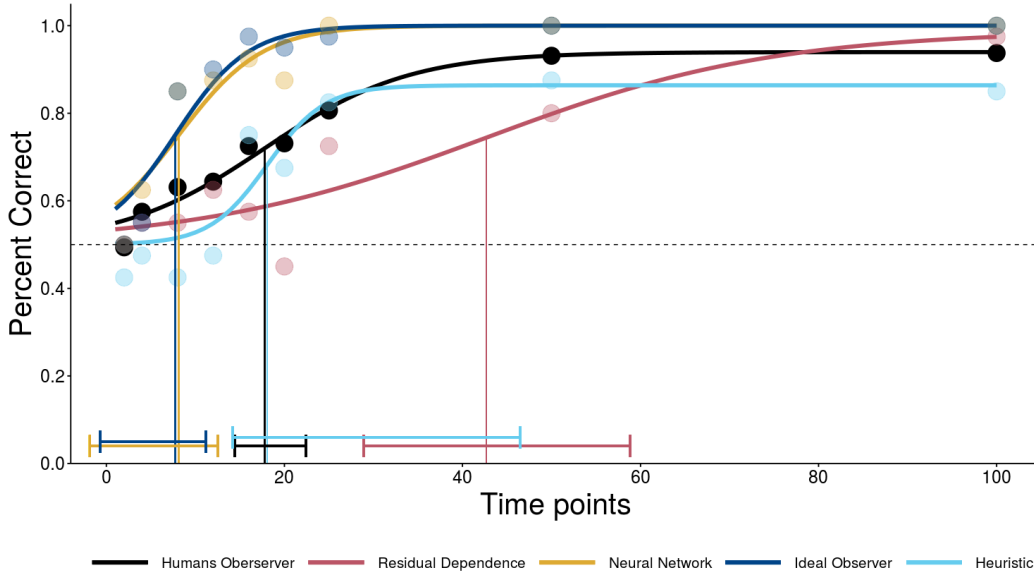

Figure 2: Performance of humans and algorithms for time series (bimodal 0.5) of different lengths, plot conventions are as in figure 1.

four of our ML algorithms. Furthermore, the graph shows that all algorithms show only a consistency consistent with them being independent. (Because we can generate more data for the algorithmic comparisons, we confirmed this using many more trials, reaching the same conclusion, see Figure A.18.). Finally we note that human observers and our heuristic show a high observed consistency.

## 4 Conclusion

Our frozen noise paradigm shows that ideal observers and the neural networks have unique strategies. Even if we use more data points in figure A.18 we see only a small effect of similarity. One could argue that we do not find an agreement of the ideal observer and neural network due to the intrinsic problem of the ideal observer algorithm for short time series. But even if we redo the frozen noise paradigm using long sequences but varying the exponent—thus rendering the sequences difficult not by shortening them but by making the noise more Gaussian—we again see only a minor effect, see figure A.19. The ideal observer and the neural network use different, albeit equally successful, strategies.

Despite the fact that, on the one hand human observers, ResDep and the heuristic and on the other hand the neural network and the ideal observer, show roughly the same performance in experiment 1, the frozen noise paradigm in experiment 2 allows us to conclude that they actually all use independent strategies. In particular, human observers do not use a ResDep dependency algorithm, and neither do they use an ideal (or suboptimal, see A.3.2) Bayesian probability calculation—especially the latter is a popular notion in visual perception and the cognitive sciences. Instead, our human observers appear to use an approach similar to our (simple) heuristic.

Another main outcome of our study is how remarkably efficient the unique strategy of the visual system is: Our observers only needed 17.76 95% CI [14.40,22.44] time points (15.17, 95% CI [11.51,19.18] if we exclude one somewhat poorer performing observer) for 75% correct discrimination of the forward or backward played AR motion sequences. They require fewer data samples than a successful ML algorithm for causal inference (ResDep; with the caveat regarding implementation mentioned above. A different implementation of the ResDep ideas may perform better). Performance approached that of the ideal observer that knows the underlying statistics perfectly, i.e., the order of the AR process, the AR coefficients, the variance and exponent of the noise of the time series. We deem it unlikely that the human observers could extract these parameters from visual input alone, let alone for the very short sequences. Our heuristic, on the other hand, is

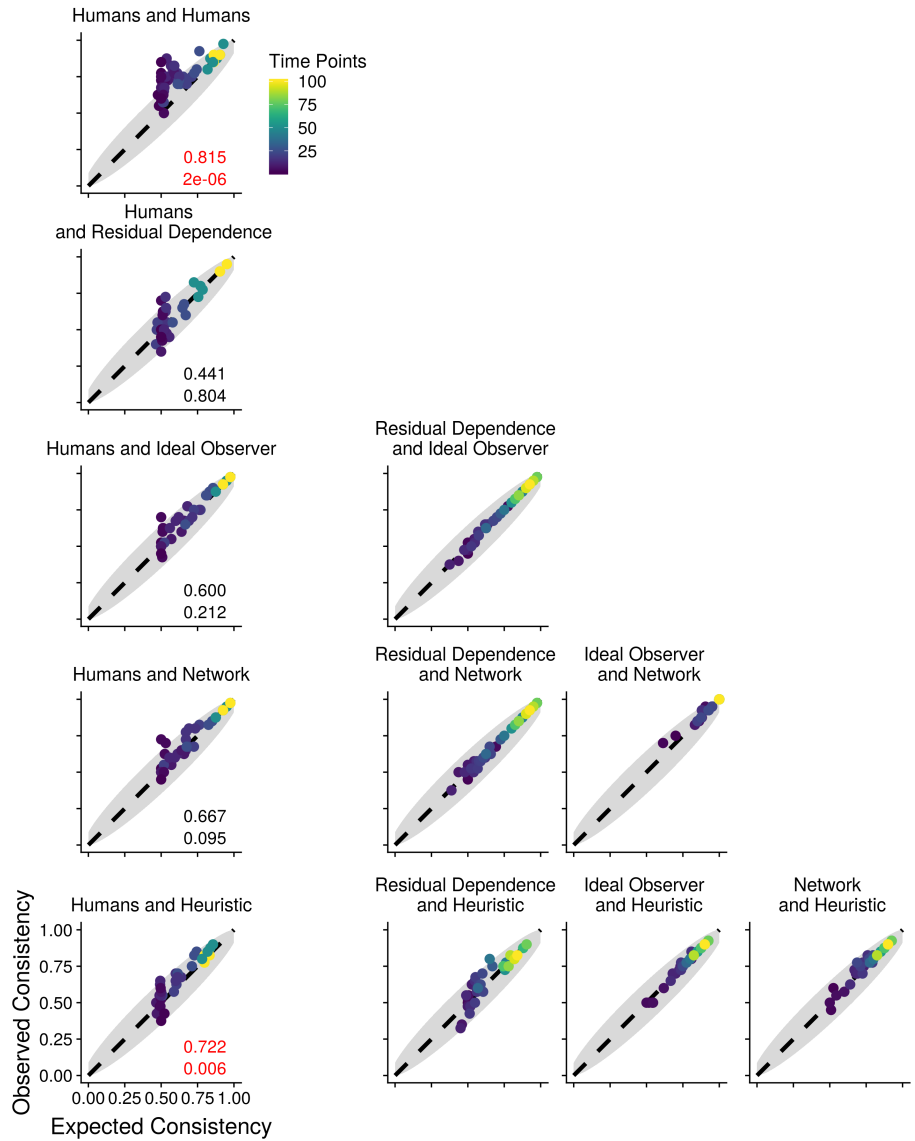

Figure 3: Human observer consistency and observer-algorithmic consistency for the frozen noise paradigm. The x-axis shows the expected proportion of equally answered trials under the assumption of independent observers or algorithms. The y-axis shows the actual observed number of equally answered trials in the experiment. The shaded area shows a 95% confidence interval calculated based on the Wilson score interval [55]. Colour codes the number of time points. We used in the algorithm-algorithm comparison not only time series with lengths from the experiment but also a finer grid: 10-30 time points with spacing 1 and 35-100 with spacing 5. The upper number on the right shows the proportion of point lying above the diagonal, the lower number the p-value for the null hypothesis that the same number of points are lying above as below the diagonal. Red numbers indicate significant deviations from the null hypothesis.

implemented in a few lines of code, is incredibly stable and fast. The surprisingly good performance and the high degree of observed consistency with humans indicates that human observers may be using a rather similar strategy.

In addition we also tried to use suboptimal algorithms [53][3]. We used two approaches to make the Bayesian observer and the Neural Network suboptimal. First, we fitted an additive noise term to the decision variable (Model 2 in Stengård and Berg). This corresponds to late noise in the visual pathway. Second, we fitted an additive noise term to the individual time series points before calculating the decision of the ideal observer. This corresponds to noise in the early visual pathway,

that is uncertainty about the exact location of the disk (e.g. micro-saccadic eye movements). The addition of both early and late noise yields more similar fits between the algorithms and human data. However, on a trial-by-trial basis, only minor similarities are visible. See section A.3.2 in the appendix for details.

When performing demanding psychophysical experiments with human observers there is always the question of learning—are we really reporting and interpreting stationary performance? In causal inference in cognitive science structure learning from data is an important topic (e.g. dynamical causal learning [10]). However, in our experiments observers were able to do the motion discrimination after a few training trials. More importantly, the accuracy in the first and second half of every block was very similar. Average performance in experiment 1 pooled for all subjects and across all noise distributions was 82% for trials 1-20 and 80% for trials 20-40; 64%/63% in experiment 2. This strongly suggests that our data are not contaminated by learning effects during our experiments.

One puzzle we are unable to resolve is why our human observers typically failed to reach above chance performance with the smoothed Uniform distribution: performance for smoothed Uniform was at 50% across all observers. From a psychophysical point of view the smoothed Uniform condition was more difficult by experimental design: Observers could not start with easy smoothed Uniform noise since there was no free parameter. On the other hand, Bimodal and smoothed Uniform distributions have a similar dependence structure, see Figure A.1. We expected that at least those observers that were already trained on Bimodal noise should be able to detect the direction of the smoothed Uniform time series—however, that was not the case. Only observer 10 achieved an accuracy above 65%. (The JS-Divergence of the smoothed Uniform distribution corresponds to a Bimodal exponent of 0.73. As we can see from Figure 1 we expect around 65% performance, in line with LL's performance.) The surprising difficulty of the smoothed Uniform distribution should help constrain which strategy or algorithm was used by our human observers during our experiments. Recent advances in causal inference have been strongly driven by human intuition about how the shape of joint distribution indicates causal directions [56, 33]. This line of argument, together with our experimental results, suggests that many of the human abilities regarding the recognition of causal and time asymmetries are not known yet.

In any case, we argue that we can learn a lot about the inner workings of a cognitive system by probing it with appropriate artificial—not ecologically valid–stimuli [57, 58] —this is not to say one should only and always use simple, artificial stimuli, but there is a place for their use, particularly when studying less well known areas—such as the human visual system's sensitivity to subtle temporal dependencies. In a predictive coding framework, e.g., it would be useful to know the exact temporal statistical structure of e.g. the motion of leaves and grass in the wind. An unusual motion pattern—e.g. having the "wrong" dependencies—may signal a hidden predator behind the foliage.

Ever since Albert Michotte performed his studies there is the question whether causal inference may under certain circumstances already be a perceptual rather than a cognitive ability [26, 28]. In our experiment observers were able to discriminate very subtle temporal asymmetries, similar to the remarkable sensitivity to higher-order spatial dependencies in patches of natural images [45]. To us this hints at an early, perceptual locus in our experiments.

**Author contributions**

B.S. had the initial project idea connecting causal inference with (early) visual perception. B.S. and F.W. developed the idea of using AR motion and additive noise as a visual stimulus with help from D.J. The concrete psychophysical experiments with all parameters were designed by K.M.; F.W. suggested the use of the "frozen noise" inspired analyses; K.M. conducted all experiments and wrote all the code. K.M. did the statistical analyses and implemented the algorithmic observers with help from F.W. and D.J. The paper was jointly written by K.M. and F.W. with input from D.J. and B.S.

**Acknowledgments**

This work was supported by the German Research Foundation (DFG): SFB1233, Robust Vision (project number 276693517): Inference Principles and Neural Mechanisms, TP4 Causal inference strategies in human vision (F.W. and B.S.).

We would like to thank Frank Jäkel for invaluable intuitions about the structure of the arrow of time problem. Additionally, we thank Heiko Schütt and Bernhard Lang for discussion about Bayesian observers and Robert Geirhos for discussion about neural networks. Moreover we are grateful to Karin Bierig and Vincent Plikat for help with data collection and Silke Gramer for administrative and Uli Wannek for technical support. Finally we thank the internal reviewers at the Max-Planck Institute for Intelligent Systems and our three anonymous reviewers for constructive feedback. We are particularly indebted to reviewer #3 and the suggestion to explore suboptimal observers.

## Footnotes

[1] https://www.youtube.com/watch?v=W_bB0TVTwg8

[2]The best three human observers for Super-Gaussian noise had thresholds of $r = 1.32, 1.36, 1.38$—at least as sensitive as ResDep.

[3]We would like to thank one of our reviewers for suggesting the suboptimal analyses

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
