[Supplementary Material]

# A  Supplementary Material

Section A.1 and A.2 provide in depth explanation and all information necessary for reproducing our experiments. Section A.3 contains information about the algorithms and the suboptimal observer analysis. Section A.4 and following show additional figures.

## A.1  Generation of time series

We constructed time series with different noise types. Table 1 lists all distributions used in constructing our time series. Figure A.1 shows the distributions' densities and the dependence of residuals in forward and backward direction for a fourth-order time series. This independence is used by the algorithm proposed by Peters et al. [37].

The (forward) time series are constructed by the following rule

$$x_t = 0.05 \cdot x_{t-4} + 0.1 \cdot x_{t-3} + 0.2 \cdot x_{t-2} + 0.4 \cdot x_{t-1} + \epsilon_t. \tag{1}$$

The first four values of $x_t$ were set to zero and the consecutive 400 time points are drooped in order to make time series stationary. The mean of all noise distributions was set to 0 and the standard deviation to 44.72 pixels on screen (1,13 cm). These values ensure that the time series is bounded to the range of possible coordinates of the monitor used in our experiment. Backward time series were constructed in the way that we first constructed new forward time series and then flipped the series along the time axis

In the Super-Gaussian and Bimodal case the noise was created according to the following rule:

$$\epsilon_t = sgn(Y) \cdot |Y|^r, \tag{2}$$

while $Y \sim \mathcal{N}(\mu,\ \sigma^2)$. We calculate, with the change of variable technique, the density of the noise as

$$p(e_t) = \frac{1}{r} \cdot |\epsilon_t|^{\frac{1}{r}-1} \frac{1}{\sqrt{2\pi}\sigma} \cdot e^{-\frac{\epsilon_t^{\frac{2}{r}}}{2\sigma^2}}. \tag{3}$$

The variance is calculated as

$$Var(\epsilon_t) = \frac{2^r}{\sqrt{\pi}} \cdot \sigma^{2r} \cdot \Gamma(r + \frac{1}{2}). \tag{4}$$

This variance of the noise depends only on $\sigma$ and the exponent $r$. While fixing the exponent, the free parameters $\sigma$ can be used to set the variance of the noise distribution to 2000.

Please note that the above distribution is not well defined for $\epsilon_t = 0$ if $r > 1$. However one can still show that the area under the distribution is still defined for $r > 1$

$$\int_0^\infty \epsilon_t^{\frac{1}{r}-1} e^{-\epsilon_t^2} d\epsilon_t = \int_0^1 \underbrace{\epsilon_t^{\frac{1}{r}-1} e^{-\epsilon_t^2}}_{<\epsilon_t^{\frac{1}{r}-1}} d\epsilon_t + \int_1^\infty \underbrace{\epsilon_t^{\frac{1}{r}-1} e^{-\epsilon_t^2}}_{<e^{-\epsilon_t^2}} d\epsilon_t < \infty. \tag{5}$$

Table 1: Noise distributions used for constructing time series and the corresponding exponents.

| Distribution | Parameter (r) |
|---|---|
| Super-Gaussian | 6 |
| Super-Gaussian | 4 |
| Super-Gaussian | 2 |
| Super-Gaussian | 1.8 |
| Super-Gaussian | 1.6 |
| Super-Gaussian | 1.41 |
| Super-Gaussian | 1.3 |
| Bimodal | 0.1 |
| Bimodal | 0.3 |
| Bimodal | 0.5 |
| Bimodal | 0.6 |
| Bimodal | 0.7 |
| Bimodal | 0.76 |
| Bimodal | 0.8 |
| Gaussian | 1 |
| smoothed Uniform | - |

Figure A.1: Noise distributions (left panel) used in our experiments and for generated time series plots of fitted residuals $\epsilon_t$ over variables $x_{t-1}$ in causal direction (middle panel) and anti causal direction (right panel). The independence in middle panel and the dependence in the right panel is used by the ResDep algorithm to determine the direction of the time series [37].

A similar approach was chosen for the smoothed Uniform distribution. The smoothed Uniform distribution has the following form

$$p(z|w,c) = \frac{1}{k} \frac{1}{1 + (\frac{z^2}{w^2})^c}. \tag{6}$$

The constant $k$ is a normalization term

$$k = \frac{\pi \cdot w}{2 \cdot c \cdot sin(\frac{\pi}{2 \cdot c})}, \tag{7}$$

the term $c$ determines the steepness of the tails and the $w$ variable controls the width of the distribution. We choose to fix the constant $c$ to 6.
One can show that for $c = 6$

$$Var(\epsilon_t) = \frac{1}{k} \frac{w^3 \pi}{3 \cdot \sqrt{2}}. \tag{8}$$

We solve this equation again for the width parameter and in the next step for the normalization parameter to equalize the variance to 2000. Values for random noises were generated with the build-in Matlab generator for the Bimodal, Super-Gaussian and Gaussian distribution. Rejection sampling was used for the smoothed Uniform distribution.

## A.2 Psychophysical Experiment

### A.2.1 Paradigm and procedure

Figure A.2 shows a static stimulus from our experiment (animated stimuli are available in the supplementary material). The position of the moving disk was directly obtained from the time series. A constant was added to centre the time series. The moving disk had a diameter of 1.5 cm, corresponding to 0.6 degrees of visual angle for the 70 cm distance between observers and display; the disk was slightly blurred with a centered Gaussian distribution of standard deviation of 0.252 cm. Vertical bars show the centre of the screen to help participants in judging the position of the dot. Stimuli were presented against a black background. The colour in normalized values for green dots was (0,1,0), for red dots was (1,0,0) during training and the bars were again (1,1,1). In "proper" experimental trials the dot colour was a neutral white (1,1,1).

Figure A.2: Static stimuli for the psychophysical experiment. In the learning trials, participants saw the moving dot with the corresponding color. In experimental trials, the color was changed to white and subjects had to press a button for the corresponding color.

Prior to the experiment written consensus was collected from all participants. Observers were told a cover story that they should imagine working as doctors and that they view bacteria through a microscope. They were told to classify the bacteria in harmless bacteria (green during the short training, [causal, term not used for participants]) and dangerous bacteria (red during the short training, [anti-causal, term not used for participants]). Neither the noise distributions nor anything about time-reversal, the arrow-of-time or causal or anti-causal directions were communicated to the participants nor any other information about the underlying scope of the experiment.
For each noise distribution (Super-Gaussian, Bimodal, smoothed Uniform, Gaussian) they saw 10 learning trials in the beginning where the dots were already colored. In case of Super-Gaussian and Bimodal, where we could change the difficulty with the exponent, the training trials were the easiest condition (r = 0.1 and r

Table 2: Ordering for stimulus presentation for the first experiment for each subject.

| Observer | Day 1 | | Day 2 | |
|---|---|---|---|---|
| | 1. | 2. | 1. | 2. |
| 1 | Bimodal | Gauss | Su-Gauss | sUnif |
| 2 | sUnif. | Bimod. | Su-Gauss. | Gauss |
| 3 | Su-Gauss. | Gauss | Bimod. | sUnif. |
| 4 | Bimod. | sUnif | Su-Gauss | Gauss |
| 5 | Su-Gauss | Gauss | Bimod. | sUnif. |
| 6 | sUnif. | Su-Gauss. | Bimod. | Gauss |
| 7 | Bimod. | sUnif | Su-Gauss | Gauss |
| 8 | Su-Gauss. | Gauss | Bimod. | sUnif. |
| 9 | Bimodal | Gauss | Su-Gauss | sUnif |
| 10 | Su-Gauss | Gauss | Bimod. | sUnif. |

= 6). In training trials dots were already colored and observers could only passively watch the movement, no responses were collected. It usually took participants only a few trials until they reported that they were able to discriminate the dots. The ordering of conditions within the Super-Gaussian and Bimodal conditions was the same as in Table 1.

In one session we presented stimuli from the long lasting distributions (Super-Gaussian, Bimodal) and one of the short lasting distributions (Gaussian, smoothed Uniform); we never started with a Gaussian noise distribution, Gaussian noise is not identifiable and we did not want to demotivate observers, see table 2.

One trial consisted of a stimulus presentation time of maximally 25 seconds (250ms per position) and an additional response time of 1.2 seconds. The experiment was self-paced, thus participants could indicate their choice with a button press at any time during stimulus presentation; stimulus presentation stopped as soon as a button was pressed. Participants were not allowed to change their choice. Trials with no response were not counted as an answer. After each trial feedback was provided by colouring the dot at the last position either green or red for causal or anti-causal time series. Feedback was provided for one second. Afterwards, we showed a 120ms black screen as inter-stimulus interval.

We calculated the number of trials according to the following rule of thumb. In psychophysical trials the binomial distribution of correct responses is often approximated with a Gaussian distribution. The one-sided 95% confidence interval for a Gaussian distribution with probability $p_0$ is:

$$p_0 + 1.96 \cdot \sqrt{\frac{p_0(1 - p_0)}{N}} \tag{9}$$

We decided that we wanted to be able discriminate chance probability $p_0 = 50\%$ from 65% with 95% confidence. This yields a sample size of 42 and we finally settled on 40 trials in our experiment per noise distribution, parameter and observer.

Every trial was randomly chosen with a probability of 50% to be a causal or an anti-causal trial. For each noise distribution and each exponent, subjects classified 40 Trials in 2 blocks of 20 trials. At the end of each block, the overall accuracy and response time was displayed. The complete experiment took 3 hours, conducted on 2 days with 1.5 hours experimentation each. After finishing the experiment participants received a debriefing.

The paradigm for experiment 2 was slightly changed. We were interested in the effect of shorter viewing times. We used Bimodal noise with exponent $r = 0.5$ since we noticed that participants could classify these time series well. On the other hand, we expected that the accuracy drops fast enough to a range where subjects showed a performance above chance level but below ceiling performance. This helped us to investigate the relationship between humans and algorithms in more detail.

At first, we familiarized subjects again with the task. Subjects observed ten times series with Bimodal noise and exponent $r = 0.1$. Afterwards, Subjects rated 10 time series with increasing difficult Bimodal noise (r = 0.1,0.3 0.5). All participants reported that they were able to classify the time series. Then we fixed the exponent $r$ to 0.5 and reduced the maximum viewing time by reducing the number of time points as indicated in table 3. For a fairer comparison between algorithms and observers, the subjects were explicitly instructed to observe the time series as long as possible unless they were really sure about the trial, see figure A.2.1.

Figure A.3: Boxplots of the reaction times for the four observers in Experiment 2, from left to right observer 11, 12, 10, 2. The dashed line shows the maximum possible reaction time.

Table 3: Ordering for stimulus presentation for second experiment where the viewing time was reduced subsequently. The seed was used to freeze the noise for all subjects. However additionally we checked that all subjects really saw the same time series.

| Number of time points | 100 | 50 | 25 | 20 | 16 | 12 | 8 | 4 | 2 |
|---|---|---|---|---|---|---|---|---|---|
| Seed | 1001 | 1002 | 1003 | 1004 | 1005 | 1006 | 1007 | 1008 | 1009 |

We encouraged the subject to observe the time series as long as possible to prevent that subjects always answered after a short viewing period. All other parameters were constant and not changed compared to experiment 1. This second experiment lasted 1.5 hours.

### A.2.2  Observers

The first experiment was piloted with two naive observers (1 male, 1 female). They showed good performance for all noise distributions. 17 naive observers participated in the first experiment (9 female, 8 male, mean age = 26y, stdev= 6.4y). The experiment is rather demanding—attention or "concentration" or willingness to perform maximally well—and we screened participants based on their performance in what we considered an "easy condition" $r = 6, 4, 2$ (Super-Gaussian) and $r = 0, 1, 0.3, 0.5$ (Bimodal). We excluded 7 of the 17 observers after the first blocks since they did not get above 67.5% performance which was needed to be significantly different from chance level. (A post-hoc and somewhat fuzzy explanation might be that for Super-Gaussian noise with large exponents "a cue event" rarely happens. Most of the time noise with small values around 0 is sampled. Thus, it was difficult for them to determine the direction. If they were not concentrating they easily missed a cue event. Thus we think that excluded observers were not in general unable to detect the causal direction (successful observers get easily >90% in these conditions) but that the seven excluded observers were not fully focusing or concentrating on the experiment.) All observers reported normal or corrected-to-normal vision and received monetary compensation and a bonus if they got an accuracy larger than 65% at Super-Gaussian (r = 1.41), Bimodal (r=0.76) and smoothed Uniform noise. The second experiment was performed by the same two

pilots (Observer 11 and 12) and 2 good observers from the first experiments, Observer 2 and 10 (2 male, 2 female, mean age = 23y, stdev =2.4y). We did not use naive observers for experiment two since we expected no effects of the debriefing in the current experiment.

### A.2.3 Apparatus

Stimuli were presented on a $22''$ VIEWPIxx LCD monitor (VPixx Technologies, Saint-Bruno, Canada) in a dark room. The monitor had a spatial resolution of $1920 \times 1200$ pixel ($484 \times 302$mm) and a refresh rate of 100 Hz. A chinrest and a headrest were used to keep the position of the head constant during the experiment. Response collecting was done with the RESPONSEPixx (VPixx Technologies, Saint-Bruno, Canada) controller. Stimulus presentation and response recordings were controlled using Psychtoolbox 3.0.12 [59, 60] in MATLAB (Release 2016a, The MathWorks, Inc., Natick, Massachusetts, U.S) along with the iShow-library (`https://zenodo.org/record/34217`) on a desktop computer (12 core CPU i7-3930K, AMD HD7970 graphics card "Tahiti" by AMD, Sunnyvale, California, United States) running Debian 9.

## A.3 Analysis

### A.3.1 Algorithms

We describe briefly our way to fit psychometric functions to the data in experiment 1 and experiment 2. Afterwards we describe in depth the 3 algorithms (Dependence, DNN and Bayes) used in comparison to human data.

In general, all analyses are done in MATLAB R2018b, most of the plots were done in R (version 3.4.2). Psychometric functions were fitted with the psychometric toolbox [54] with a fixed guessing rate of 0.5. For experiment 1 a cumulative Gaussian function was used. We fitted a logistic function to the data of Experiment 2. No further changes to the default options were applied.

We used the ResDep algorithm proposed and offered by Bauer et al. [38] to detect the time series based on independence relationships between data and (fitted) residuals. We modified the original algorithm to run in Matlab 2018 since the *vgxset* and *vgxvarx* functions are not supported in the newest MATLAB version. To compare the algorithm to the human performance, we also used a forced-choice paradigm, thus we always force the algorithm to choose the direction in which the residuals are more independent. The algorithm outputs 0 if it fails to fit a time series. We subtracted the mean and divided by the standard deviation each time series as preprocessing step since MATLAB fittings have problems with time series of large variance. We made sure that centering the time series does not change the relationship of the residuals to the data.
For the second experiment, we slightly further modified the original algorithm. In the original algorithm, the order of the fitted time series is chosen with the Akaike Information Criterion [61] between 1 and 10. Instead of the original implementation, we fit only up to order 1 and 5. We changed the maximum fitted order since the Matlab routine has a problem to fit a short time series with long lags. The algorithm starts guessing for time series shorter or equal than 8 time steps because fitting series is not possible and additionally also the independence test does not work anymore.

The neural network was implemented in Matlab with the Deep Learning Toolbox (Version 12.0). Each Network was trained separately for each noise distribution. We generated 30,000-time series, half of it causal ones, the other half anti-causal ones. We used 75% as training data and 25% for validation. The first layer consisted of a convolutional layer of kernel size $1 \times 10$ and 10 kernels in total, followed by a batch-Normalization layer, a relu-layer, a fully connected layer and finally a softmax layer for classification. The initial learning rate was set to 0.01 together with the Adam-solver of MATLAB. We limited learning to a maximum of 30 epochs (approximately 3 minutes) and set the "Validation Patience" to 5 epochs. All other values remained to the default values. For every noise, we trained the network 3 times and chose the one with the best performance. This had only very small effects on performance.
For experiment 2 minor modifications were necessary. We reduced the number of time points $n$ until subjects only saw two points in total. Thus we had to change the architecture for the convolutional layer and changed the size of the first convolutional layer to $min(10, n)$.

The last algorithm described is the ideal observer algorithm. We calculated the probability that the time series is in causal direction ($d = 1$) or in anti causal direction ($d = 0$) given the data $X_{t=\{1,...,N\}}$

$$p(d = 1|X_{t=\{1,...,N\}}) = \frac{p(X_{t=\{1,...,N\}}|d = 1) \cdot p(d = 1)}{p(X_{t=\{1,...,N\}})} = \frac{\Pi_{t=1}^{N} p(x_t|x_{t-1},...,x_1) \cdot p(d = 1)}{p(X_{t=\{1,...,N\}})}. \quad (10)$$

since we only use time series of order p = 4 we could limit this to

$$p(d = 1|X_{t=\{1,\dots,N\}}) = \frac{p(X_{t=1,\dots,N}|d=1) \cdot p(d=1)}{p(X_{t=\{1,\dots,N\}})} \qquad (11)$$

$$= \frac{\Pi_{t=1}^N p(x_t|x_{t-1}, x_{t-2}, x_{t-3}, x_{t-4}) \cdot p(d=1)}{p(X_{t=\{1,\dots,N\}})} \qquad (12)$$

$$= \Pi_{t=5}^N p(x_t|x_{t-1}, x_{t-2}, x_{t-3}, x_{t-4}) \cdot$$
$$\frac{p(x_4|x_3, x_2, x_1) \cdot p(x_3|x_2, x_1) \cdot p(x_2|x_1) \cdot p(x_1) \cdot p(d=1)}{p(X_{t=\{1,\dots,N\}})} \qquad (13)$$

In the anti-causal direction we get

$$p(d = 0|X_{t=\{1,\dots,N\}}) = \Pi_{t=1}^{N-4} p(x_t|x_{t+1}, x_{t+2}, x_{t+3}, x_{t+4}) \cdot$$
$$\frac{p(x_{N-3}|x_{N-2}, x_{N-1}, x_N) \cdot p(x_{N-2}|x_{N-1}, x_N) \cdot p(x_{N-1}|x_N) \cdot p(x_N) \cdot p(d=0)}{p(X_{t=\{1,\dots,N\}})}$$
$$(14)$$

It follows that under assumption of equal probability of forward and backward time series that

$$\frac{p(d = 1|X_{t=\{1,\dots,N\}})}{p(d = 0|X_{t=\{1,\dots,N\}})}) \approx \frac{\Pi_{t=5}^N p(x_t|x_{t-1}, x_{t-2}, x_{t-3}, x_{t-4}) \cdot p(x_1)}{\Pi_{t=1}^{N-4} p(x_t|x_{t+1}, x_{t+2}, x_{t+3}, x_{t+4}) \cdot p(x_n)}. \qquad (15)$$

This equation holds only approximately since we skipped the terms after the first expression in eq. (13) and eq. (14). For long time series we omit these terms since they only influence the final ratio marginally. However, this could explain why the Bayes ideal observer algorithms performs worse than a Neural Network in experiment 2 for very small numbers of time points, because for short time series the approximation does not hold anymore. The Bayesian ideal observer algorithm always outputs 0 for time series smaller than 5 data points.

As earlier noted we have fitted an (ecological valid) heuristic to the data in spirit similar to heuristic presented by Stengård and Berg [53]. The heuristics were developed after we had evaluated the feedback from our subjects and the analyse of the noise structure. We found two different principles for Bimodal and Super-Gaussian noise. Informal speaking for Super-Gaussian noise, noise values are often sampled around 0. Therefore in the forward direction, the dot often jumps around the centre and rarely makes a big jump outwards. After such a big jump, the point slowly sprints back to the centre. This means that in the forward direction there are big jumps to the outside, in the backward direction there are big jumps to the centre. The Bimodal condition behaves the other way around. Often larger values are sampled and only rarely smaller ones.

We used a few lines of code to implement this heuristic. The heuristic searches for the maximum displacement and takes the maximum jump distance $n$ steps before and after the maximum. For Super-Gaussian noise we say that the time series in the forward direction if $max_{before} > max_{after}$ and for Bimodal noise if $max_{before} < max_{after}$. (If one wouldn't know the noise distribution one could use the kurtosis of fitted residuals to determine which rule to use). The number of steps $n$ is hyperparameter which can be tuned. In general, we did not find a very strong performance dependence on this parameter. In experiment 1 and experiment 2 a value of $n = 4$ was used.

Finally, we show that our heuristic also works on real data. We used the same data as [37] and determined if we should use the Super-Gaussian or Bimodal heuristic by the kurtosis of the fitted residuals. We get with a stepsize of 12 an accuracy of $60.4\%(p < 10^{-10})$for 1180 time series.

Please note that the kurtosis for all time series was larger than 3 such that we decided to use the Super-Gaussian rule. However, with this rule we only get an accuracy of 39.57%. But if we force the residual dependence algorithm to classify the exact same time series (something [37] did not do in their paper) we yield an accuracy of 42%. These EEG time series seem to some intrinsic properties which make it difficult to use these for an arrow of time detection. Thus, we report 1 minus the accuracy for our heuristic.

### A.3.2  Suboptimal Algorithms

Thanks to the suggestion of our reviewers we also tried to use suboptimal algorithms [53] to make the algorithms more similar to human performance. We used two approaches to make the Bayesian observer and the Neural Network suboptimal in experiment 2. First, we fitted a noise term additive to the decision variable (Model 2 in Stengård and Berg). This corresponds to very late (decision) noise in the visual pathway.

For this purpose we sampled 10000 time series for each length of the time series. Then we added Gaussian noise with zero mean and a specific standard deviation to the decision variable $des$

$$des = \frac{p(d = 1|X)}{p(d = 0|X)} + \eta, \ \eta \sim \mathcal{N}(\mu, \sigma_l^2). \qquad (16)$$

Afterwards we computed the accuracy for the different length of the time series and calculated the deviance between human data and the accuracy of the suboptimal algorithm [62]. A grid search between standard deviations of 0 and 20 was performed to search for the variance which yielded closes human performance. For the ideal observer a standard deviation of 2.5 was selected and for the neural network a standard deviation of 5.5 Finally we calculated performance for the data used in experiment 2 by sampling noise with the above variance and add these to the decision variables. Figure A.4 and A.5 shows the psychometric function and the correlation analysis. As expected, the psychometric functions of the ideal observer and neural network become more similar to human psychometric functions. However, the correlation analysis shows that they still classify the time series differently (non significant consistency on a single trial basis).

Second, we fitted an additive noise term to the individual time series before calculating the decision of the ideal observer.

$$x_t = x_t + \eta, \ \eta \sim \mathcal{N}(\mu, \sigma_e^2). \tag{17}$$

This corresponds to noise in the early visual pathway (uncertainty about the exact location of the disk e.g. due to micro-saccadic eye movements). The procedure equals the procedure for late noise, with the exception that the grid search was done in the range of 0 to 40. Standard deviations of 21 and 19 were selected. Figure A.6 and A.7 plots the psychometric function and the correlation analysis for early noise. The results are similar to the late noise case. Although psychometric functions become more similar we do not see a lot of agreement in the consistency analysis.

It is worth to note, that the exact curve of the psychometric function depends on the samples noise variables. We repeated this step several times to check the implications. However the overall trend and the non-consistency was in a very similar range.

Figure A.4: Psychometric functions for non optimal late noise algorithms in experiment 2. Plot conventions as in Figure 1.

Figure A.5: Consistency analysis for late noise algorithms in experiment 2. Plot conventions as in Figure 3. Black Boxes indicate the use of suboptimal algorithms

Figure A.6: Psychometric functions for non optimal early noise algorithms in experiment 2. Plot conventions as in Figure 1.

Figure A.7: Consistency analysis for early noise algorithms in experiment 2. Plot conventions as in Figure 3. Black Boxes indicate the use of suboptimal algorithms.

## A.4   Individual Results for Observers in Experiment 1

Figure A.8: Accuracy of the ten single observers in experiment 1 across all noise distributions and pooled accuracy for all humans and the algorithms. From left to right we show performance for increasing difficult Bimodal noise(b), Super-Gaussian noise(sg), Gaussian noise(g) and smoothed Uniform(su) noise. The number corresponds to the exponent of the distribution. The horizontal line shows the chance performance of 50%.

Figure A.9: Psychometric Functions of the ten single observers in experiment 1 for time series with Bimodal noise. Blacks dot represent human mean performance. Vertical lines represent the 75% threshold. Whisker represent the 95% Credible Interval.

Figure A.10: Psychometric Functions of the ten single observers in experiment 1 for time series with Super-Gaussian noise. Blacks dot represent human mean performance. Vertical lines represent the 75% threshold. Whisker represent the 95% Credible Interval.

Table 4: Thresholds and 95% Credible Interval for the 10 individual observers in experiment 1 from psychometric functions fitted by Psignifit 4[54]

| Observer | Threshold Bimodal | Threshold Super-Gaussian |
|----------|-------------------|--------------------------|
| 1 | 0.60 [0.53,0.69] | 1.66 [1.38,2.27] |
| 2 | 0.56 [0.41,0.77] | 2.19 [1.70,2.74] |
| 3 | 0.69 [0.61,0.84] | 1.36 [0.04,3.24] |
| 4 | 0.64 [0.55,0.74] | 1.68 [1.48,2.01] |
| 5 | 0.72 [0.65,0.86] | 1.58 [0.81,2.66] |
| 6 | 0.74 [0.63,0.86] | 1.32 [0.35,1.90] |
| 7 | 0.41 [0.29,0.66] | 2.41 [1.88,4.83] |
| 8 | 0.65 [0.57,0.90] | 1.70 [1.16,2.27] |
| 9 | 0.73 [0.67,0.83] | 1.74 [1.40,3.03] |
| 10 | 0.73 [0.61,0.86] | 1.38 [0.77,1.55] |

## A.5 Analysis of the neural network

Figure A.5 shows a further performance investigation of the neural network. Learned weights are shown in figure A.12 and figure A.13. We use the trained networks and tested the accuracy on all other noise distributions. The neural network generalizes better from difficult noise (r =0.8 or r=1.3) to easy noise (r=0.1 or r=6) than in the other direction. Furthermore, the network trained on Super-Gaussian noise or Bimodal noise interchange the label if we test for the other noise distribution. This effect is indicated by a performance below chance level and in extreme cases even accuracy of 0%. Bimodal noise and smoothed Uniform noise seem to show similar performance as expected from the residual distribution, see discussion in the previous section. Remarkably, some subjects told us after the experiment that when we switched the noise distributions from day 1 to day 2 they also thought that we interchanged the labels of the time series. Additionally, the network is able to capture the noise distributions when trained on Super-Gaussian and Bimodal noise simultaneously. The Super-Gaussian and Bimodal distribution had the same KL-Divergence to a Gaussian with same mean and variance.

| Trained on \ Tested on | Super-Gaussian 6 | Super-Gaussian 4 | Super-Gaussian 2 | Super-Gaussian 1.8 | Super-Gaussian 1.6 | Super-Gaussian 1.41 | Super-Gaussian 1.3 | Bimodal 0.1 | Bimodal 0.3 | Bimodal 0.5 | Bimodal 0.6 | Bimodal 0.7 | Bimodal 0.76 | Bimodal 0.8 | Gaussian | Smoothed Uniform | Mixed |
|---|---|---|---|---|---|---|---|---|---|---|---|---|---|---|---|---|---|
| Super-Gaussian 6 | 100 | 100 | 99.8 | 98.6 | 96.7 | 92.5 | 83.6 | 0 | 0.1 | 0.4 | 2.2 | 7.4 | 13.5 | 19.6 | 51.2 | 13.3 | 49.1 |
| Super-Gaussian 4 | 100 | 100 | 100 | 99.9 | 99.4 | 96.6 | 89.8 | 0 | 0 | 0.2 | 0.5 | 2.3 | 6.1 | 11.6 | 50.7 | 7.9 | 49.4 |
| Super-Gaussian 2 | 99.9 | 100 | 99.9 | 99.4 | 98.5 | 93.4 | 87.1 | 0 | 0 | 0.6 | 1.6 | 5.9 | 12.3 | 16.9 | 50 | 9.6 | 48.8 |
| Super-Gaussian 1.8 | 100 | 100 | 99.9 | 99.8 | 99.4 | 96.7 | 90.7 | 0 | 0 | 0.1 | 0.7 | 2.8 | 7 | 11.7 | 48.1 | 6.5 | 49.2 |
| Super-Gaussian 1.6 | 100 | 100 | 99.8 | 99.8 | 99.2 | 96.4 | 90.6 | 0 | 0 | 0.1 | 0.7 | 1.8 | 7.4 | 13.7 | 49.3 | 6.6 | 49.2 |
| Super-Gaussian 1.41 | 100 | 100 | 100 | 99.9 | 99.6 | 96.2 | 91.3 | 0 | 0 | 0 | 0.2 | 1.5 | 5.4 | 9.8 | 50 | 6.8 | 49.7 |
| Super-Gaussian 1.3 | 100 | 100 | 100 | 99.9 | 99.4 | 98 | 91.8 | 0 | 0 | 0 | 0.5 | 1.8 | 4.6 | 10.4 | 51.1 | 6.8 | 49.8 |
| Bimodal 0.1 | 3 | 2.2 | 5.1 | 8.6 | 9.2 | 17.5 | 22.9 | 100 | 99.9 | 99 | 94.7 | 87 | 82.1 | 75.7 | 50.3 | 82.4 | 54.1 |
| Bimodal 0.3 | 0.9 | 0 | 0.3 | 1.2 | 3 | 6.4 | 14.1 | 100 | 100 | 99.8 | 99.2 | 94.3 | 90.1 | 84 | 48.8 | 90.5 | 51.3 |
| Bimodal 0.5 | 0.1 | 0 | 0 | 0.3 | 0.6 | 2.2 | 8.7 | 100 | 100 | 99.9 | 99.7 | 98.5 | 94.6 | 89 | 48.3 | 92.4 | 50.6 |
| Bimodal 0.6 | 0.2 | 0 | 0.1 | 0.7 | 1.6 | 5 | 11.6 | 100 | 100 | 99.9 | 99.5 | 95.5 | 92.3 | 85.8 | 51.8 | 92.1 | 51.1 |
| Bimodal 0.7 | 0.1 | 0 | 0 | 0.3 | 0.1 | 3.4 | 9 | 100 | 100 | 99.9 | 99.7 | 98.6 | 94.7 | 88.6 | 50.5 | 94.1 | 50.5 |
| Bimodal 0.76 | 0.3 | 0 | 0 | 0 | 0.7 | 3.3 | 8.1 | 100 | 100 | 100 | 99.8 | 99.1 | 95.5 | 90.1 | 51 | 93 | 50.9 |
| Bimodal 0.8 | 62.1 | 36.1 | 11.3 | 7.7 | 8.1 | 11.6 | 16.8 | 100 | 100 | 100 | 99.1 | 95.9 | 91.1 | 83.7 | 49.4 | 95.8 | 54.1 |
| Gaussian | 49.1 | 49.4 | 50.7 | 49.1 | 48 | 46 | 50.2 | 51.2 | 53.2 | 51.4 | 49 | 48.9 | 50 | 50.6 | 51.5 | 52 | 52.5 |
| Smoothed Uniform | 48.2 | 25.2 | 5.1 | 4.7 | 5.4 | 9.8 | 16.9 | 100 | 100 | 100 | 99 | 95.7 | 90.2 | 84.7 | 50.2 | 96.3 | 54 |
| Mixed | 79.3 | 86.8 | 81.3 | 76.9 | 69.9 | 65.2 | 59 | 99.2 | 99 | 93.8 | 86.4 | 71.9 | 62.2 | 58.8 | 50.2 | 68.5 | 77 |

Acc. 100 / 50 / 0

Figure A.11: Generalization of the Neural Network across different noise distributions. We trained the network on one noise distribution(rows) and tested against all other distributions (columns). The mixed condition consists of a dataset where half of the time series have Bimodal noise and the other Super-Gaussian noise with equal KL divergence compared to Gaussian distribution with same mean and variance.

**Super-Gaussian 6**

**Super-Gaussian 4**

**Super-Gaussian 2**

**Super-Gaussian 1.8**

**Super-Gaussian 1.6**

**Super-Gaussian 1.41**

**Super-Gaussian 1.3**

**Bimodal 0.1**

**Bimodal 0.3**

**Bimodal 0.5**

**Bimodal 0.6**

**Bimodal 0.7**

**Bimodal 0.76**

**Bimodal 0.8**

**gaussian**

**smootheduniform**

**mixed**

Figure A.12: Weights of the first convolutional layer of the neural network across all noise distributions. We used 10 kernels of length 10. The weights for the easier tasks are (exponents away from 1) are in general smaller than weights for more difficult cases (weights close to 1).

Figure A.13: Weights of the first convolutional layer of the neural network across all noise distributions. Black lines show the weights for output units in causal direction and grey lines the weights for output units in anti-causal direction.

## A.6 Performance of human observers in other distance spaces

The parameterization with exponent $r$ is somewhat arbitrary and not directly linked to a parameterized difficulty-scale. A more natural parameterization could be the distance between noise distribution and the non-identifiable Gaussian distribution with the same mean and variance. We calculate the distance with two Information Theory-based f-divergences (Kulback-Leiber Divergence and the symmetric Jensen-Shannon Divergence) as well as the Kolmogorov–Smirnov statistic in figure A.14. Human psychometric functions overlap for the exponent in the JS-space and overlap less in the KL-divergence or KS-distance space. Thus, the performance of humans seems to be captured rather well if we express the distributional distances in the JS-divergence space. Furthermore, if we plot the Bimodal psychometric function on a $1/r$ scale, the difficulty of the Bimodal and Super-Gaussian conditions for human observers is roughly equal, indicating that in our parameterized difficulty for human observers is reasonably well captured by the distance of $1/r$ for the Bimodal and $r$ for the Super-Gaussian noise.

Figure A.14: Psychometric functions of human observers in other distance spaces. The first plot shows the psychometric function where we converted the exponents to a KL-divergence between the noise distribution and a Gaussian distribution with the same mean and variance. The second plot the conversion to the symmetric Jenson-Shanon divergence and the third plot the Kolmogorov-Smirnov distance. The last plot show the psychometric function when we inverted all Bimodal exponents (r<1). We scaled the x-axis for the JS-Divergence, KS-Divergence and Exponents such that all Bimodal psychometric functions have the same slope, scale values are 3.5 (JS), 2.91 (KS), 0.296 (Exponents). Thus we can compare the threshold distances graphically. Vertical lines show thresholds and whiskers show the 95% Credible Interval for thresholds from Psignifit.

## A.7    Block by block comparison of experiment 1

The psychometric functions summarize performance as a function of a single independent variable. To understand and compare human observers and the different algorithms it is sometimes instructive to compare the performance of humans and algorithms on a block-by-block basis (40 trials per block), as shown in Figure A.15. Each data point represents the performance of one of the 10 observers for one noise distribution on the x-axis plotted against algorithmic performance for the same time series on the y-axis; in addition we compare the performance of the algorithms to each other. The ResDep algorithm is a little better than humans (almost 60% of blocks are above the diagonal). The ideal observer and the neural network are, not surprisingly given figure 1 better for almost every block of 40 trials than the human observers and there is little correlation between human observers and the two algorithms. Additionally, the heuristic has similar performance as humans and the residual dependence algorithm.

Figure A.15: Performance comparison on a block basis between humans and ResDep, ideal observer and neural network. Every single symbol corresponds to one block with 40 trials. Different symbols correspond to different noise distributions.

Figure A.16: Individual performance of observers in experiment 4. The vertical line indicates chance performance. All 4 observers show a similar performance even for very short time series. Individual Thresholds are for Subject 11: 14.45, 95% CI [6.29 22.35], Subject 12: 14.33, 95% CI [7.53,20.86], Subject 10: 19.04, 95% CI [10.58,30.07], Subject 2: 30.55.04, 95% CI [20.53,58.48].

## A.9 Results of experiment 2 for 3 Observers

Figure A.17: Human observer consistency and observer-algorithmic consistency for the frozen noise paradigm. Plot conventions as in figure 3 but excluded poor performing subject BW.

## A.10 Consistency analysis for algorithms with more samples

Figure A.18: Revaluation of lower left part in Figure 3 with more trials. Expected consistency versus observed consistency for algorithms for 1000 trials per time point condition. Lengths of time series ranged from 5-31 time point with spacing 1 and 35-100 with spacing 5.

## A.11 Frozen noise analysis between ideal observer and network for difficult Super-Gaussian noise

Figure A.19: Ideal observer and network consistency for the frozen noise paradigm. Similar to figure A.18. But we avoid the intrinsic problems of the ideal observer algorithm 2.3 with shot time series by using again time series with 100 time points and made the time series more difficult by making the exponent closer to 1 (Gaussian noise). 1000 times series per exponent were used. The x-axis shows the expected proportion of equally answered trials under the assumption of independent observers or algorithms. The y-axis shows the actual observed number of equally answered trials in the experiment. Shaded area shows a 95% confidence interval calculated based on the Wilson score interval [55]. Color codes the Exponent. We used 30 exponents in the linear range between 1.1 and 1.4.