[Reviews · NeurIPS 2019]

Reviewer 1



Originality: To the best of my knowledge, conducting human psychophysics and comparing their performance to computational models has not been done for the precise problem formulation examined by the authors. However, the authors did not describe previous work on anorthoscopic perception, which has examined similar question (how do people perceive a figure that is revealed to them through a slit moving over it over time? Is the constructed perception equivalent in each order?) and has a long history dating back to Helmholtz. For a good review, see Rock, I. (1981). Anorthoscopic Perception. Scientific American, 244(3), 145-153 (https://www.researchgate.net/profile/Marc_TESSERA/post/Is_mathematics_a_human_contrivance_or_is_it_innate_to_nature/attachment/59d62f0ec49f478072e9f719/AS%3A273518934069250%401442223403218/download/1981AnorthoscopicPerceptionRock.pdf). Quality: The human experiments and computational modeling are well conducted. Very high quality. My main criticism of the computational modeling is that the neural network that was tested gets different evidence than that given to participants. I would have preferred to see a comparison to a simple recurrent neural network that only gets subsequences in an online fashion rather than one that gets the entire sequence at once. Clarity: Given the amount of material and the page limits, the paper is clear. It is well-written. There are some, minor experimental details that I think should be in the article itself. I will list them here for the authors (they did not affect my score, although I would expect some of them to be updated). Line 117: Units given in pixels. This is device-dependent. Please provide a device-independent measurement. You could add in resolution and screen size or inches. Lines 136-137: How were they blocked. Did they have to be \geq 67.5% accuracy for each block? Lines: 142-144: Were there learning effects? Was it the same noise per trial randomly sampled once and then the same precise sequence is given to all participants/use the same seed or was the noise randomly generated for each trial and subject? Former is preferred, but regardless, please make clear. More general: Were participants told the noise distribution? What cover story were they given? Significance: Generally, causal perception and reasoning is a very, important problem of high interest to the NeurIPS community. I am sympathetic to the authors for examining a tractable problem that is similar in difficulty for people and models. However, it has low ecological validity. This would be less of an issue if the authors explained human performance using one or more of the computational models. However, it seems like people act differently from all of the models presented. **Author feedback response: I thank the authors for responding to my criticism. I agree with them that "toy" stimuli are important for uncovering basic mechanisms. However, even these toy stimuli typically have some resemblance to a real-world task (e.g., direction in a random dot motion task is similar to what small noisy observations of a moving surface might look like to a perceptual system). I believe the connection in the task mentioned by the authors in their response is weak. I still think the paper should be accepted and published. My hope is that the authors take the feedback to guide future work and perhaps that it may be worthwhile to revise some of the motivation for the manuscript.

Reviewer 2



Reads great. Finally someone tests this.

Reviewer 3



After reading the Author Feedback: The authors fully addressed all my concerns, and even provided preliminary fits for "suboptimal" Bayesian observers (to be fully analyzed in the final revision), which show better agreement with human performance, at least in terms of accuracy. Looking forward to the "frozen noise" analysis. Overall, this is a very interesting and well-thought manuscript, and I am happy to increase my score from 7 to 8. Summary: In this paper, the authors investigate whether humans are able to distinguish the "arrow of time" of a simple perceptual stimulus (a moving dot) which moves in time according to an autoregressive time series with non-Gaussian noise. In a first experiment, the authors investigate human performance in discriminating "forward" from "backward" motion as a function of the non-Gaussianity of the noise distribution (mostly, taken from a family parametrized by the exponent of a normal random variate). In a second experiment, the authors use a "frozen noise" approach and sequences of different lengths to investigate more in detail the strategies used by the human participants. Human performance is compared with three candidate algorithms (neural networks, a Bayesian ideal observer and an existing "residual dependence" algorithm), both in terms of psychometric curves (and estimated psychometric thresholds) and via a correlation analysis. In particular, the "frozen noise" approach allows the authors to see how human participants and different algorithms perform on the same exact sequences, and provides a means to judge the consistency between these strategies. Originality: This is an original study in both its question, experimental setup, and methods of analysis. Quality: The quality of this paper is high, with attention to detail from the experimental design to the careful statistical analysis. A couple of times, I was wondering about some detail or control check, and the manuscript would provide a detailed, thoughtful analysis that would answer my question soon after. Clarity: The main text is very well written and clear. Significance: This is an important contribution to the field, with interesting experimental techniques and analyses that can bear impact to future studies. Major comments: This is a methodologically sound, interesting and well-written paper. I mostly have comments about the Bayesian ideal observer modeling. First, when writing the posterior ratio used by the observer to determine whether the series is causal or anti-causal, the authors drop a few leading/ending terms (Eq. 14 in the Supplementary Material). This approximation is not completely inconsequential - as the authors themselves note that "However, this [approximation] could explain why the Bayes ideal observer algorithms performs for a small number of time points worse than a Neural Network in experiment 2. For short time series the approximation does not hold anymore. The Bayesian ideal observer algorithm always outputs 0 for time series smaller than 5 data points." Given these issues, it's not clear to me why this approximation is even necessary in the first place; as it looks like the correct equation could be easily implemented in a few lines of extra code (perhaps ugly, but it shouldn't be a big deal in terms of tractability). The only potential difficulty I see, unless I am missing some other detail, is an assumption about the initial distribution, p(x_1). However, there is a natural choice, that is the stationary distribution of the time series (or, perhaps, for simplicity, a Gaussian approximation thereof) convolved with the noise. Second, it's unlikely that observers would behave as perfectly ideal Bayesian observer (i.e., know the noise distribution perfectly; have perfect memory; compute the posterior ratio perfectly; judge stimulus position exactly at each time step, etc.). There are several ways to make an observer sub-optimal (or boundedly rational -- and perhaps closer to human behavior), from noisy sensory observations to decision noise. Understandably, a full (e.g., factorial) analysis of all these model components is beyond the scope of this paper and left for future work, but at least I would expect some discussion of it. If the authors feel like fitting a suboptimal Bayesian observer model to the data, a recent work (Stengård & van den Berg, 2019) has shown that different forms of noise can be approximated well by "decision noise", so this is a suboptimal observer that might be worth exploring first. On a technical point, how did the authors obtain "frozen noise"? The supplement reports some "seed" values (Table 3 in Supplementary Material) which I imagine correspond to the seed passed to a rng() function in Matlab. However, to my own disappointment, seeds are not guaranteed to produce the same stream between different machines, even with the same operating system and Matlab version, so this is not enough for reproducibility and also may create issues if the authors used different machines to run the experiments/analyses (unless a full sequence of random numbers was separately stored and passed around). I also went through the Supplementary Material, which is very detailed and I congratulate the authors for making all this material and additional analyses available. One minor note though - it is noticeable that the Supplement was written somewhat hastily, as there is a substantially larger number of typos, and the style is occasionally unclear. I would recommend the authors to revise it for resubmission. Finally, I strongly encourage the authors to make their code available (both for running the experiment and for the analysis). Minor comments: line 117: "These values ensure that the time series is bounded to the range of possible coordinates of the monitor used in our experiment." Maybe I am missing something, but it's not clear to me how these values *ensure* that the time series is bounded - perhaps the authors mean that these values make it extremely unlikely that the time series would go outside the bounds? (unless the authors explicitly added a truncation somewhere). Maybe an "in practice" is sufficient here to clarify. line 225: How is the neural network trained for Experiment 2 - it's explained in the Supplementary Information but perhaps it's better to mention it also again in the main text - I understand that the network is retrained from scratch separately for each sequence duration? Typos: line 22: networks(CNNs) --> networks (CNNs) line 261: if use --> if we use line 290: 0.63% --> 63% (I hope) Supplementary Material: Eq. 3: Check the symbols (e.g., e_t should be \epsilon_t, x and y?) Eq. 4: asterisks should be \cdot line 462: Rejecting --> Rejection line 490: Trails --> Trials line 508: change --> chance Table 3: Please explain a bit more in detail what the Seed number refers to here (see main comments) line 519-535: This section is somewhat hard to parse, perhaps revise for clarity. line 533: model --> bimodal? line 536: Apperatus --> Apparatus line 574: validation patience --> ??? line 582: sine --> since line 586: incidentally --> marginally? line 619: 40 blocks per trial --> 40 trials per block? References: Stengård, E., & Van den Berg, R. (2019). Imperfect Bayesian inference in visual perception. PLoS computational biology, 15(4), e1006465.

[Author Response · NeurIPS 2019]

We would like to thank all reviewers for their valuable feedback and we very much appreciate their assessment of our work as "very high quality" (R1), "top 15% of accepted NeurIPS papers" (R2) and "well-thought experimental setup" (R3) in studying causal inference in humans. We will make sure to address all the minor issues raised by the reviewers in the final version of our paper. In the following we reply individually to the main issues raised:

**R1:** *I would like to see some explanation of how people solve the task using the models and ecological validity.* We agree that our visual system has not evolved to discriminate "in-time" from time-reversed events. However, we can learn a lot about the inner workings of a cognitive system by probing it with *appropriate* artificial—not ecologically valid–stimuli (RUST & MOVSHON, *Nature Neuroscience*, 2005; MARTINEZ-GARCIA ET AL, *Frontiers in Neuroscience*, 2019)—this is not to say we should *only* use simple, artificial stimuli, but there is a place for their use, particularly when studying less well known areas—such as the human visual system's sensitivity to subtle temporal dependencies. In a predictive coding framework, e.g., it would be useful to know the exact temporal statistical structure of e.g. the motion of leaves and grass in the wind. An unusual motion pattern—e.g. having the "wrong" dependencies—may signal a hidden predator behind the foliage. Thus whilst we agree with reviewer R1 that the exact experiment of a moving single disk lacks ecological validity, the implications do not—we see this as the beginning of explorations into subtle temporal dependency structures in biological motion in general, causal inference abilities, leader-follower behaviour etc. Finally, we fitted three (!) more ecological valid models to the data, see discussion of R3's comments and the figure. Obviously we will attempt to make our reasoning clearer in the revised version of the manuscript.

**R1:** *It's not clear to me why the particular time-series and/or the noise parameterization are generalizable to new situations or special in some other manner*: It is only possible to classify time series in our setting with non-Gaussian noise. One measure for non-Gaussianty is kurtosis. Our parameterization yields Gaussian, platykurtic (bimodal) as well as leptokurtic (super-Gaussian) noise from the same equation. Thus, we belief we cover a broad range of non-Gaussian noise, and we can control the degree of non-Gaussianity.

**R1:** *I would have preferred to see a comparison to a simple recurrent neural network.* R1 is of course correct that the network gets as input the entire sequence which is different to humans. However, the network starts with a convolutional layer of size 10 which effectively slides over the time series. Thus we think that despite the seemingly very different nature of the inputs, de facto our scenario is somewhere in between a sequential RNN and a fully connected layer in which the network could use the evidence of the entire time series as a first step.

**R2:** Thank you very much for your review, we highly appreciate it.

**R3:** *Fitting a suboptimal Bayesian observer model to the data*: Thank you for pointing out this excellent paper to us. We agree that this is indeed a very promising way to extend our work. Based on the proposed paper, we test three additional strategies, shown in the figure on the right (dashed line is the ideal observer from the original submission). First, we fitted a noise term additive to the decision variable (Model 2 in Stengård and Berg). This corresponds to late noise in the visual pathway. Second, we fitted an additive noise term to the individual time series before calculating the decision of the ideal observer. This corresponds to noise in the early visual pathway and uncertainty about the exact location of the disk (e.g. micro-saccadic eye movements). Third, we have fitted a (ecological valid) heuristic to the data in spirit similar to

those proposed by Stengård and Berg. This heuristic needs only a few lines of code, but it yields results close to that shown by our observers. We will add a discussion and comparison of the three new algorithms to our paper, in particular with an analysis of their consistency using frozen noise (Fig. 3 of the original submission).

**R3:** *Approximation used for the Bayesian likelihood distribution* We agree that it would be better to use the full likelihood. Estimating the stationary distribution $p(x_1)$ is good advice which we will of course implement. In addition we will assess if and how we can estimate or approximate the other 3 conditional distributions given that the time series values are not independent, we skip the first 400 terms and numerical estimation of the multidimensional (conditional) distributions could be challenging.

**R3:** *On a technical point, how did the authors obtain "frozen noise"?:* As you thought, we fixed the seed for the random number generator, but in addition—as a cautionary measure, as we worry the way you do—we saved all time series which were generated during the experiments. Thus we can confirm (ensure) that all observers and algorithms classified exactly the same time series.

[Meta-Review · NeurIPS 2019]

The reviewers are quite consistent in liking this paper. I hope that some of reviewer 1's comments can be addressed in revision.